

# Getting the most out of RNA-seq data analysis

Tsung Fei Khang[1] and Ching Yee Lau[2]

[1] Institute of Mathematical Sciences, University of Malaya, Kuala Lumpur, Malaysia
[2] Institute of Biological Sciences, University of Malaya, Kuala Lumpur, Malaysia

Corresponding author
Tsung Fei Khang,
tfkhang@um.edu.my

## ABSTRACT

**Background.** A common research goal in transcriptome projects is to find genes that are differentially expressed in different phenotype classes. Biologists might wish to validate such gene candidates experimentally, or use them for downstream systems biology analysis. Producing a coherent differential gene expression analysis from RNA-seq count data requires an understanding of how numerous sources of variation such as the replicate size, the hypothesized biological effect size, and the specific method for making differential expression calls interact. We believe an explicit demonstration of such interactions in real RNA-seq data sets is of practical interest to biologists.

**Results.** Using two large public RNA-seq data sets—one representing strong, and another mild, biological effect size—we simulated different replicate size scenarios, and tested the performance of several commonly-used methods for calling differentially expressed genes in each of them. We found that, when biological effect size was mild, RNA-seq experiments should focus on experimental validation of differentially expressed gene candidates. Importantly, at least triplicates must be used, and the differentially expressed genes should be called using methods with high positive predictive value (PPV), such as NOISeq or GFOLD. In contrast, when biological effect size was strong, differentially expressed genes mined from unreplicated experiments using NOISeq, ASC and GFOLD had between 30 to 50% mean PPV, an increase of more than 30-fold compared to the cases of mild biological effect size. Among methods with good PPV performance, having triplicates or more substantially improved mean PPV to over 90% for GFOLD, 60% for DESeq2, 50% for NOISeq, and 30% for edgeR. At a replicate size of six, we found DESeq2 and edgeR to be reasonable methods for calling differentially expressed genes at systems level analysis, as their PPV and sensitivity trade-off were superior to the other methods'.

**Conclusion.** When biological effect size is weak, systems level investigation is not possible using RNAseq data, and no meaningful result can be obtained in unreplicated experiments. Nonetheless, NOISeq or GFOLD may yield limited numbers of gene candidates with good validation potential, when triplicates or more are available. When biological effect size is strong, NOISeq and GFOLD are effective tools for detecting differentially expressed genes in unreplicated RNA-seq experiments for qPCR validation. When triplicates or more are available, GFOLD is a sharp tool for identifying high confidence differentially expressed genes for targeted qPCR validation; for downstream systems level analysis, combined results from DESeq2 and edgeR are useful.

## INTRODUCTION

Elucidating key genes associated with variation between different biological states at the genomic level typically begins with the mining of high dimensional gene expression data for differentially expressed genes (DEG). For a long time, biologists have been using microarrays for gene expression studies, and over the years, the collective experience of the community has congealed into a set of best practices for mining microarray data (*Allison et al., 2006*). Hence, to determine optimal replicate size, one may use the SAM package (*Tibshirani, 2006*); to call DEG, the moderated *t*-test (*Smyth, 2005*; *Smyth, 2004*) would be applied (*Jeanmougin et al., 2014*), producing *p*-values for each gene that adjust for multiple comparisons (*Dudoit, Shaffer & Boldrick, 2014*). Subsequently, joint consideration of *p*-value and fold change (*Xiao et al., 2014*) allows the researcher to identify a set of DEG with strong potential to be validated by qPCR. Riding on such confidence, the researcher could further study functional enrichment to gain understanding of dysregulated biological processes, or generate network-based hypotheses for targeted intervention.

Despite the microarray's analytical maturity, RNA-seq, which is based on next-generation sequencing technology—is set to become the method of choice for current and future gene expression studies (*Wang, Gerstein & Snyder, 2009*). In RNA-seq, direct transcript counting through mapping of short reads to the genome overcomes the problem of limited dynamic range caused by signal saturation in microarrays. In addition, the transcriptome can now be sequenced to unprecedented coverage, thus removing dependence on prior transcriptome knowledge which is crucial for probe design in microarrays. With the availability of numerous *de novo* transcriptome assembly tools (*Li et al., 2014*), meaningful gene expression studies in non-model organisms can now be done. While conceptually simple, RNA-seq requires the use of sophisticated algorithms to transform raw reads into the final gene counts. These algorithms constitute an important source of non-biological variation that must be appropriately accounted for *Oshlack, Robinson & Young (2010)*, if the result is to be interpretable.

The limited availability of biological material, and the costs of data production and bioinformatic support, are the major limiting factors for replicate size in RNA-seq experiments. As a result, RNA-seq data sets with little or no biological replicates remain quite common today. In these type of data sets, statistical power to detect DEG is poor, and further deteriorates when the biological effect size is not strong. In fact, the problem may become worse from a multiple comparison point of view, as potentially many more genes are scored. Studies that aim at a systems level understanding using the list of DEG must therefore prioritize large replicate sizes over sequencing depth (*Rapaport et al., 2013*). However, large RNA-seq experiments remain the exception, rather than the rule at the moment.

The count-based nature of RNA-seq data prompted new development of statistical methods to call DEG. Despite the latter, differential gene expression analysis remains

challenging due to lack of standard guidelines for experimental design, read processing, normalization and statistical analysis (*Auer & Doerge, 2010*; *Auer, Srivastava & Doerge, 2012*). Currently, there is a bewildering number of methods for calling DEG. Several recent studies compared the relative performance of various DEG call methods using simulated and also real RNA-seq data sets (eleven in *Soneson & Delorenzi (2013)*; five in *Guo et al. (2013)*; eight in *Seyednasrollah, Laiho & Elo (2015)*). Recommendations for method selection were offered. However, these studies did not explicitly consider variation of the performance of DEG call methods in the context biological effect size and unreplicated experiments, which are of practical concern to the biologist. It may not be an overstatement to say that, at present, how researchers pick a DEG call method out of the plethora of alternatives available is more guided by their degree of familiarity with the methodology literature, computing convenience and democratic evaluation of personal experiences in bioinformatics forums, rather than on empirical evidence.

Most DEG call methods are designed to address analysis of RNA-seq experiments that have biological replicates. A minority such as ASC (*Wu et al., 2010*), NOISeq (*Tarazona et al., 2011*) and GFOLD (*Feng et al., 2012*) were initially designed for analysis of unreplicated experiments, though the latter two could also handle replicated experiments. While unreplicated experiments are not suitable for reliable inference at the systems level, DEG mined using particular DEG call methods may nonetheless be useful for targeted study if their expression can be validated independently using qPCR. Such small incremental gains can be crucial to build up the ground work in preparation for more extensive study in non-model organisms. Our study aims to clarify the interaction between replicate size, biological effect size and DEG call method, so as to provide practical recommendations for RNA-seq data analysis that will help researchers get the most out of their RNA-seq experiments.

## MATERIALS AND METHODS

### Statistical methods for calling differentially expressed gene

A large number of DEG call methods have been proposed (Table 1), with the majority of them being parametric methods that make distributional assumption about the read count data. An exhaustive comparison of all available methods for the present study was not feasible, nor necessary, since the relative performance of various subsets of these methods have been investigated in several studies (*Soneson & Delorenzi, 2013*; *Guo et al., 2013*; *Zhang et al., 2014*; *Seyednasrollah, Laiho & Elo, 2015*). As a result, less promising methods can be omitted from comparison.

Comparisons involving methods specifically designed for unreplicated experiments received little attention, despite the abundance of such type of RNA-seq data. For this reason, we included ASC, GFOLD and NOISeq. For replicated experiments, we focused on methods that have received the most attention from the scientific community (as reflected by their relatively high citations per year), such as edgeR, DESeq and its new version, DESeq2. These are parametric methods that explicitly model the distribution of count data using the negative binomial distribution. Initially designed for standard

**Table 1 Methods for calling differentially expressed genes in RNA-seq data analysis.** Total citations were based on Google Scholar search result as of 22 September 2015, and normalized by number of years since formal publication. The methods were ranked according to their citations per year.

| Method | Total citations | Citations per year | Reference |
|---|---|---|---|
| DESeq[*] | 2,987 | 597 | *Anders & Huber (2010)* |
| edgeR[*] | 2,260 | 452 | *Robinson, McCarthy & Smyth (2010)* |
| Cuffdiff2 | 517 | 258 | *Trapnell et al. (2013)* |
| DESeq2[*] | 209 | 209 | *Love, Huber & Anders (2014)* |
| voom[*] | 143 | 143 | *Law et al. (2014)* |
| DEGseq | 592 | 118 | *Wang et al. (2010)* |
| NOISeq[*,a,b] | 324 | 81 | *Tarazona et al. (2011)* |
| baySeq | 310 | 62 | *Hardcastle & Kelly (2010)* |
| SAMSeq[b] | 114 | 57 | *Li & Tibshirani (2013)* |
| EBSeq | 107 | 53 | *Leng et al. (2013)* |
| PoissonSeq | 99 | 33 | *Li et al. (2012)* |
| BitSeq | 70 | 23 | *Glaus, Honkela & Rattray (2012)* |
| DSS | 46 | 23 | *Wu, Wang & Wu (2013)* |
| TSPM | 70 | 17 | *Auer & Doerge (2011)* |
| GPseq | 86 | 17 | *Srivastava & Chen (2010)* |
| NBPSeq | 65 | 16 | *Di et al. (2011)* |
| QuasiSeq | 47 | 16 | *Lund et al. (2012)* |
| GFOLD[*,a] | 44 | 15 | *Feng et al. (2012)* |
| ShrinkSeq | 30 | 15 | *Van De Wiel et al. (2013)* |
| NPEBseq[b] | 14 | 7 | *Bi & Davuluri (2013)* |
| ASC[*,a] | 32 | 6 | *Wu et al. (2010)* |
| BADGE | 2 | 1 | *Gu et al. (2014)* |

**Notes.**

[*] Methods included in the present study.

[a] Methods initially developed to analyze unreplicated RNA-seq data sets.

[b] Non-parametric method.

Programming language: C/C++ for GFOLD, Cuffdiff2 and BitSeq; Matlab for BADGE; R for the rest.

experiments with biological replicates, these methods were later modified to accommodate analysis of unreplicated experiments as well, but their performance relative to ASC, GFOLD and NOISeq remains unclear. We did not include two methods with high citations per year: Cuffdiff2 and DEGSeq, based on conclusions from recent method comparative analyses. For example, Cuffdiff2 was found to have very low precision when replicate size increased in the analysis of two large RNA-seq data sets from mouse and human (*Seyednasrollah, Laiho & Elo, 2015*). Furthermore, *Zhang et al. (2014)* showed that edgeR had slightly superior performance in the receiver operating characteristic curve compared to DESeq and Cuffdiff2. Another comparative study involving DESeq, DEGseq, edgeR, NBPSeq, TSPM and baySeq showed that DEGseq had the largest false positive rate among them (*Guo et al., 2013*).

The inclusion of the popular non-parametric method NOISeq provides a contrast between performance of parametric and non-parametric methods. We included voom, which connects log-transformed read count data to the mature limma analysis pipeline

**Table 2 Description of the core modeling strategy of differential gene expression analysis methods investigated in the present study.**

| Method | Description | Reference |
|---|---|---|
| NOISeq | Non-parametric modeling of odds of signal against noise; NOISeqBIO is a variant for handling replicated experiments which integrates the non-parametric framework of NOISeq with an empirical Bayes approach | Tarazona et al. (2011) Tarazona et al. (2015) |
| ASC | Empirical Bayes shrinkage estimation of log fold change | Wu et al. (2010) |
| GFOLD | Poisson count distribution; Bayesian posterior distribution for log fold change | Feng et al. (2012) |
| edgeR | Negative binomial count distribution; genewise dispersion parameter estimation via conditional maximum likelihood; empirical Bayes shrinkage of dispersion parameter; exact test for $p$-value computation | Robinson, McCarthy & Smyth (2010) |
| DESeq | Negative binomial count distribution; local regression modeling of mean and variance parameters | Anders & Huber (2010) |
| DESeq2 | Negative binomial count distribution; generalized linear model; shrinkage estimation of dispersion parameter and fold change | Love, Huber & Anders (2014) |
| voom | Estimates of mean–variance trend from log-transformed count data are used as input for the limma empirical Bayes analysis pipeline developed for microarray data analysis | Law et al. (2014) |
| $Z$-test | The $Z$-statistic for testing the equality of two proportions | – |

(*Smyth, 2004*; *Smyth, 2005*) that has been used so successfully for detecting DEG in microarray data analysis. Finally, the $Z$-test for equality of two proportions was included to set upper bounds in the tested performance metrics that are attainable by naive application of a common textbook statistical method. Let $N_{ij}$ be the pooled normalized read counts of the $i$th gene in the $j$th phenotype class ($j = 1, 2$), $N_{+j} = \sum_i N_{ij}$ the total normalized read counts in the $j$th phenotype class, and $N_{i+} = \sum_{j=1,2} N_{ij}$ the total normalized read counts of the $i$th gene in all phenotype classes. Specifically, the $Z$-test statistic for the $i$th gene is given by

$$Z_i = \frac{\hat{p}_{i1} - \hat{p}_{i2}}{\sqrt{\hat{p}_i(1 - \hat{p}_i)/N}},$$

where $\hat{p}_{ij} = N_{ij}/N_{+j}$, $\hat{p}_i = N_{i+}/N$, and $N$ is the total number of normalized counts. Table 2 provides a description of the core modeling approaches of the eight methods considered in the present study.

### Criteria for differential expression

For edgeR, DESeq, DESeq2 and $Z$-test, we used a joint filtering criteria (*Li, 2012*) based on fold change ($\phi$) and $p$-value ($p$) to call DEG. Let $y = -\log_{10} p$ and $x = \log_2 \phi$. Thus, each gene is associated with a paired score $(x, y)$ after differential expression analysis. Following (*Xiao et al., 2014*), we required $p < 0.01$ and $\phi \geq 2$ to call for up-regulated genes, and $p < 0.01$ and $\phi \leq 1/2$ to call for down-regulated genes. The product of $y > 2$ and

$|x| \geq 1$ yields the inequality $y > 2/|x|$. Thus, genes that fell in the region defined by $y > 2/x$ were differentially up-regulated, and those in the region of $y > -2/x$ were differentially down-regulated. The union of the sets of differentially up and down-regulated genes constituted the set of DEG candidates.

To handle analysis of unreplicated experiments in edgeR, we set the biological coefficient of variation (BCV) parameter as 0.4 for the Cheung data set (see details in 'Benchmarking'), and 0.1 for the Bottomly data set, following recommendations in *Chen et al. (2015)*. The exact test option was used to compute *p*-values.

For NOISeq, we used the recommended criteria for calling DEG as described in the NOISeq documentation—$q = 0.9$ for unreplicated experiments, and $q = 0.95$ for experiments with biological replicates. For ASC, genes that had log2 fold change above 1 or $-1$, and posterior probability 99% or more were declared to be differentially expressed. For GFOLD, we used the default significant cut-off of 0.01. A gene with GFOLD value of 1 or larger was considered differentially up-regulated, and differentially down-regulated if GFOLD value was $-1$ or smaller. Except GFOLD, which is written in the C/C++ language and requires the Linux platform, the other methods were executed in R version 3.1.3 (*R Core Team, 2015*).

## Benchmarking

### Data sets

To set up our benchmarking exercise, we needed two RNA-seq data sets whereby variation in their phenotype classes produced mild and strong biological effect sizes in the tissue of interest, respectively. We further required the RNA-seq data sets to have fairly large replicate sizes to enable the simulation of different replicate size scenarios. To this end, we identified two suitable RNA-seq data sets in the Recount database (*Frazee, Langmead & Leek, 2011*). The latter contains unnormalized RNA-seq count data sets from 18 major studies that have been assembled from raw reads using the Myrna (*Langmead, Hansen & Leek, 2010*) pipeline.

The Bottomly data set (*Bottomly et al., 2011*) consists of gene expression data (22 million Illumina reads per sample, read length of ∼30 bases) obtained from the brain striatum tissues of two mice strains: C57BL/6J ($n = 10$) and DBA/2J ($n = 11$). Both mice strains are known to show large, strain-specific variation in neurological response when subjected to opiate drug treatment (*Korostynski et al., 2006*; *Korostynski et al., 2007*; *Grice et al., 2007*).

The Cheung data set (*Cheung et al., 2010*) consists of gene expression data (40 million Illumina reads per sample, read length of 50 bases) from immortalized human B-cells of 24 males and 17 females. Sex hormones are known to modulate B cell function (*Klein, 2000*; *Verthelyi, 2001*). For example, estrogen modulates B cell apoptosis and activation (*Grimaldi et al., 2002*), while testosterone suppresses immunoglobulin production by B cells (*Kanda, Tsuchida & Tamaki, 1996*). In the absence of antigenic challenge, however, it seems reasonable to expect only a modest number of DEG in male and female B cells.

After removal of transcripts with zero counts in all samples, the Bottomly count table contained 13,932 transcripts, down from an initial 36,536 transcripts, whereas the Cheung

count table contained 12,410 transcripts, down from 52,580. Prior to analysis, the count data were normalized using DESeq normalization (*Anders & Huber, 2010*), which has been shown to be robust to library size and composition variation (*Dillies et al., 2013*). However, raw counts were used for DESeq2 analysis since the method explicitly requires such type of data as input.

### Method for constructing a reliable reference DEG set

The construction of a reliable reference DEG set from which performance metrics for each method is evaluated is a non-trivial problem, if one eschews a simulation-based approach. To avoid circular reasoning, this reference set needs external validation from independent evidence such as confirmation from qPCR results.

Here, we chose voom (*Law et al., 2014*; *Ritchie et al., 2015*) as the method of choice for setting the reference DEG set. Unlike other DEG methods that primarily model mean–variance relationships in the count data using discrete distributions such as the Poisson or negative binomial distributions, voom log-transforms count data into a microarray-like data type suitable for analysis using the robust limma pipeline (*Smyth, 2004*; *Ritchie et al., 2015*). Because of this, using voom to set the reference DEG set can avoid biasing results of the called DEG due to algorithmic similarities. A gene was defined as differentially expressed using the same joint filtering criteria for edgeR, DESeq, DESeq2 and $Z$-test. We found the nonparametric SAMSeq (*Li & Tibshirani, 2013*), which has also been reported to have strong DEG mining performance, unsuitable for setting the reference DEG set as it returned different DEG sets for different random seeds and number of permutation parameters (Fig. S1).

The validity of voom as a tool for constructing reasonable *in silico* reference DEG sets for the Bottomly and Cheung data set requires justification. To this end, we compared its performance with other DEG call methods on an RNA-seq data set in which qPCR validation results for sufficiently large numbers of genes are available. Briefly, the Rajkumar data set (*Rajkumar et al., 2015*) consists of gene expression count data (26,119 genes; minimum of 10 million Illumina reads per sample, read length of ∼50 bases) from the amygdala tissues of C57BL/6NTac strain mice. There are two phenotype classes: wild type ($n = 8$), and heterozygotes for the *Brd1* gene deletion ($n = 8$). A total of 115 genes were selected for qPCR validation (additional Table 5 in *Rajkumar et al., 2015*); differential expression was observed in 60 of them, and not in the remaining 55. Each DEG call method returns $N_g$ differentially expressed gene candidates. We considered a method to be sound for setting the reference DEG set if it did not return too few (tens) or too many (thousands) candidates. Among methods that satisfied this criterion, the one that had relatively higher positive predictive value (PPV; the complement of the false discovery rate) would be preferred. Let $N_{TP}$ be the number of true positives, and $N_{FP}$ the number of false positives. Then the number of DEG that lack validation result is $U = N_g - N_{FP} - N_{TP}$. If $U$ is not too large or too small, then the expected number of true positives can be estimated as

$$N_{TP}^* = N_{TP} + \frac{N_{TP}}{N_{TP} + N_{FP}} U.$$

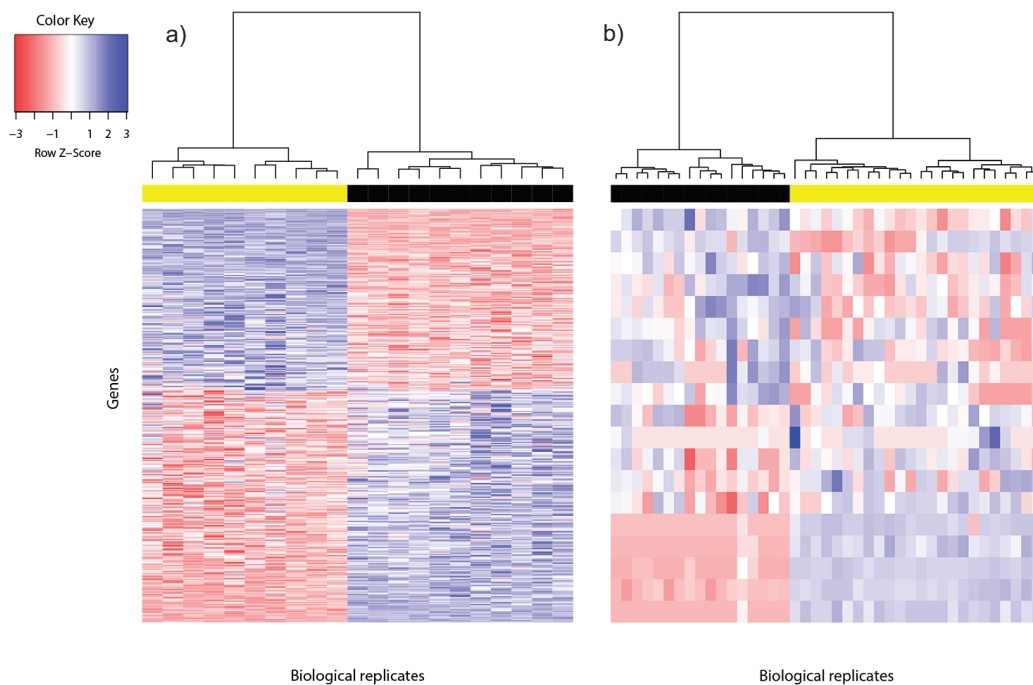

**Figure 1 Heat map of differentially expressed genes in (A) Bottomly data set (362 DEG) and (B) Cheung data set (19 DEG).** Phenotype class legend: (A) Black for DBA/2J strain ($n = 11$); yellow for C57BL/6J strain ($n = 10$). (B) Black for female ($n = 17$); yellow for male ($n = 24$). The heat maps were made using the `gplots` (*Warnes et al., 2014*) R package. Pairwise sample distances were estimated using the Euclidean distance and sample clustering was done using the Ward algorithm. The DEG were sorted based on the magnitude and sign of their $t$-statistic.

The expected PPV is therefore given by

$$PPV^* = \frac{N_{TP} + N_{TP}^*}{N_g}.$$

### Characteristics of constructed reference DEG sets

Ideally, the *in silico* reference DEG set called using voom for the two test data sets should be independently validated using qPCR, but evidence at such level may not always be available. Where microarray data are available for the same study, a DEG candidate can be considered reliable if it is called in both RNA-seq and microarray analyses, since fold change of DEG from the latter has been found to correlate strongly with fold change from qPCR (*Wang et al., 2014*). A total of 362 DEG for the Bottomly data set were thus called (Fig. 1A). About 88% (320/362) of the DEG for the Bottomly data called using voom were identical with those called in *Bottomly et al. (2011)* using edgeR (1,727 DEG). Approximately two fifths of them (153/362) were detected using limma applied on Affymetrix or Illumina microarray expression data (Table S1).

For the Cheung data set, gender difference was the source of phenotype class variation. We exploited this fundamental biological difference to infer the most reliable DEG from the candidates returned using voom. Only DEG which were located on the sex

chromosomes, or interacted with at least one gene product from the sex chromosomes were used to construct the reference DEG set. This strategy resulted in a set of 19 DEG (Fig. 1B). Five of them were located on the Y chromosome, three on the X chromosome and the remainder had known gene-gene interactions with at least one gene located on sex chromosomes (based on BioGRID (*Stark et al., 2006*; *Chatr-Aryamontri et al., 2015*); Table S2).

Differentially expressed genes are characterized by between-phenotype variation that is significantly larger than within-phenotype variation. However, occasionally some genes may be wrongly declared as differentially expressed because some outliers within a phenotype class were sufficiently extreme to cause relatively large between-phenotype variation. To assess the quality of DEG called using voom, we used the Bland-Altman plot (*Bland & Altman, 1986*; Fig. S2). Among the 362 DEG called for the Bottomly data set, the majority of DEG showed good agreement of replicate variation between the two phenotype classes—about 88% (318/362) were within 2SD (standard deviation) from perfect agreement, and about 95% (345/362) were within 3SD. Similarly, among the 19 DEG called for the Cheung data set, within-phenotype variation difference was within 2SD from perfect agreement for about 74% (14/19) of DEG, and within 3SD for about 84% (16/19) of DEG. Generally, genes that showed large within-phenotype variation in both phenotypes were not called by voom.

Once the DEG set had been constructed for the Bottomly and Cheung sets, it became possible to operationally define what we meant by mild or strong biological effect size. For the $i$th differentially expressed gene, define

$$T_i^2 = \frac{(\bar{X}_{i,1} - \bar{X}_{i,2})^2}{S_{i,1}^2/n_1 + S_{i,2}^2/n_2},$$

where $i$ indexes the genes, and $j$ the phenotype classes ($j = 1, 2$); $\bar{X}_{i,j}$ and $S_{i,j}^2$ are the mean and variance of normalized read counts respectively, and $n_j$ are the replicate sizes. Thus, $T^2$ is essentially the square of the $t$-statistic, which measures the magnitude of squared deviation between mean counts in two different phenotype classes relative to the latters' variances. By definition, the median values of $T^2$ should be large in a data set that shows strong biological effect size, and vice versa. For the Bottomly data set (strong effect size), median $T^2$ was 27.6; for the Cheung data set (mild effect size), it was 4.6. Both data sets had approximately equal spread of $T^2$ values around the median, the interquartile range being 38.3 and 34.5 for the Bottomly and Cheung data sets, respectively.

## Simulation and performance evaluation

To simulate unreplicated experiments in both data sets, we used all possible sample pairs ($11 \times 10 = 110$) for the Bottomly data set, and 300 random sample pairs for the Cheung data set. Except ASC, which only handles unreplicated experiments, we further examined the behavior of other DEG call methods in cases of low to modest number of replicates. We constructed 100 instances of experiments for each replicate size per phenotype class in the Cheung data set ($n = 3, 6, 10$), and the Bottomly data set ($n = 3, 6$) by random sampling without replacement within each phenotype class.

**Table 3  DEG set size and expected PPV of the DEG call methods in the analysis of Rajkumar data set.** Variation in DEG set size and expected PPV were computed using bootstrapping for methods where the DEG set size was not too small or too large.

| Method | DEG set size ± SE | *PPV** ± SE (%) |
|---|---|---|
| voom | 287 ± 43 | 88.9 ± 4.1 |
| edgeR | 564 ± 694 | 72.6 ± 15.0 |
| DESeq | 3,384 | NR |
| *Z*-test | 9,417 | NR |
| DESeq2 | 10 | NR |
| NOISeq | 31 | NR |
| GFOLD | 38 | NR |

**Notes.**

SE, standard error; NR, Not Relevant.

To evaluate method performance, we used sensitivity and positive predictive value (PPV). For each DEG call method, we computed sensitivity as the proportion of reference DEG that were called. PPV was computed as the proportion of DEG called that were members of the reference DEG set. The mean and standard deviation (SD) of these metrics were then reported. Methods that show good PPV are particularly interesting in the context of unreplicated experiments, since DEG obtained from them offer the best potential of being validated. For systems level analysis, DEG should preferably be called using methods with good balance of sensitivity and PPV.

# RESULTS & DISCUSSION

## Validity of voom for setting the reference DEG set

The DEG set size and expected PPV of each method in the analysis of the Rajkumar data set are given in Table 3. The results indicate that only voom and edgeR produced call sizes that were of reasonable order of magnitude. However, voom had relatively higher expected PPV over edgeR; additionally, the DEG set size called using voom had standard error (SE) that was an order of magnitude smaller compared to edgeR (bootstrap sampling with replacement of biological replicates; 1,000 iterations). Therefore, it seemed reasonable to use voom as the method of choice to construct the reference DEG set for the Bottomly and Cheung data sets.

We note with interest from Table 3 that the number of DEG called by DESeq2 dropped drastically compared to DESeq. Since DESeq2 implements a shrinkage estimation of dispersion parameter and fold change to improve the performance of DESeq, the present suggests that this may occasionally lead to over-correction, resulting in DEG set size that is too small.

## Performance of DEG call methods in the Cheung and Bottomly data sets
### *Positive predictive value and sensitivity*

The ASC package provided by *Wu et al. (2010)* failed to run for particular combinations of sample pairs. Only 24.5% (27/110) and 41.3% (124/300) pairs of samples from the
Bottomly and Cheung data set respectively could analyzed using ASC. The simulation results show that optimality of a DEG call method for a given replicate size depended on whether biological effect size was mild or strong (Fig. 2). In the Cheung data set (mild biological effect size), all methods had very low (about 1%) mean positive predictive value (PPV) for unreplicated experiment (Fig. 2A and Table S3), suggesting that no meaningful biological insights were possible. However, mean PPV ($\pm$SD) increased substantially for NOISeq to $43.5 \pm 31.5\%$, and for GFOLD to $29.6 \pm 15.8\%$, for $n = 3$ (Fig. 2B). Doubling and approximately tripling replicate size to $n = 6$ (Fig. 2C) and $n = 10$ (Fig. S3) further improved mean PPV for NOISeq to $87.0 \pm 16.1\%$ and $92.2 \pm 12.9\%$, and for GFOLD to $36.3 \pm 14.9\%$ and $52.6 \pm 18.8\%$, respectively. In all four replicate sizes, mean PPV was low for the other methods. It did not exceed 12% for DESeq2, and was never more than 3% for edgeR, DESeq and $Z$-test.

A markedly different pattern of method performance was observed in the analysis of the Bottomly data set (strong biological effect size). In unreplicated experiments (Fig. 2D), mean PPV was relatively high for NOISeq ($50.6 \pm 20.3\%$), ASC ($47.2 \pm 25.9\%$) and GFOLD ($31.2 \pm 25.6\%$), compared to just about 15% in edgeR and 5% in DESeq and $Z$-test. DESeq2 did not perform well, with mean PPV ($29.6 \pm 28.4\%$), and an extremely low sensitivity ($0.2 \pm 0.6\%$) as a result of making too few calls. Interestingly, GFOLD attained very high mean PPV at $n = 3$ ($94.3 \pm 6.9\%$; Fig. 2E), with marginal change to $92.5 \pm 3.3\%$ at $n = 6$. However, GFOLD was also the method with the lowest sensitivity (below 10%) for these two replicate sizes, which was caused by its small DEG set size (Fig. 3).

DESeq2 struck the best balance between PPV and sensitivity as replicate size increased, but edgeR showed reasonable performance too. At $n = 3$ (Fig. 2E) and $n = 6$ (Fig. 2F), DESeq2 had mean PPV of $52.5 \pm 10.8\%$ and $62.1 \pm 7.7\%$, with mean sensitivity of $36.0 \pm 5.7\%$ and $65.1 \pm 4.5\%$, respectively. For edgeR, its mean PPV was $28.7 \pm 4.1\%$ and $33.9 \pm 3.0\%$, with mean sensitivity of $59.8 \pm 5.4\%$ and $79.0 \pm 4.6\%$, respectively. At $n = 6$, DESeq2 had similar sensitivity compared to its older version DESeq, and a superior mean PPV that was about four times higher. Unsurprisingly, the $Z$-test remained the worst performer, with mean PPV just about 6%.

The general increase in mean sensitivity for replicated experiments was consistent with the finding that the increase in statistical power for detecting DEG is primarily determined by biological replicate size, and less by sequencing depth (*Liu, Zhou & White, 2014*).

### DEG set size

Figure 3 shows the distribution of DEG set size in the Cheung and Bottomly data sets for different replicate sizes (for details, see Table S3). Although DESeq2 could be used to call DEG for unreplicated experiments, it was shown to behave erratically in the Bottomly data set, with extremely low mean DEG set size ($2.5 \pm 6.8$). In general, for replicated studies, methods such as DESeq2, DESeq, edgeR and $Z$-test made large numbers of calls that were typically one or two order of magnitudes more (depending on underlying biological effect size) compared to GFOLD or NOISeq. Consequently, it is expected that the formers' sensitivity would increase at the expense of their PPV.

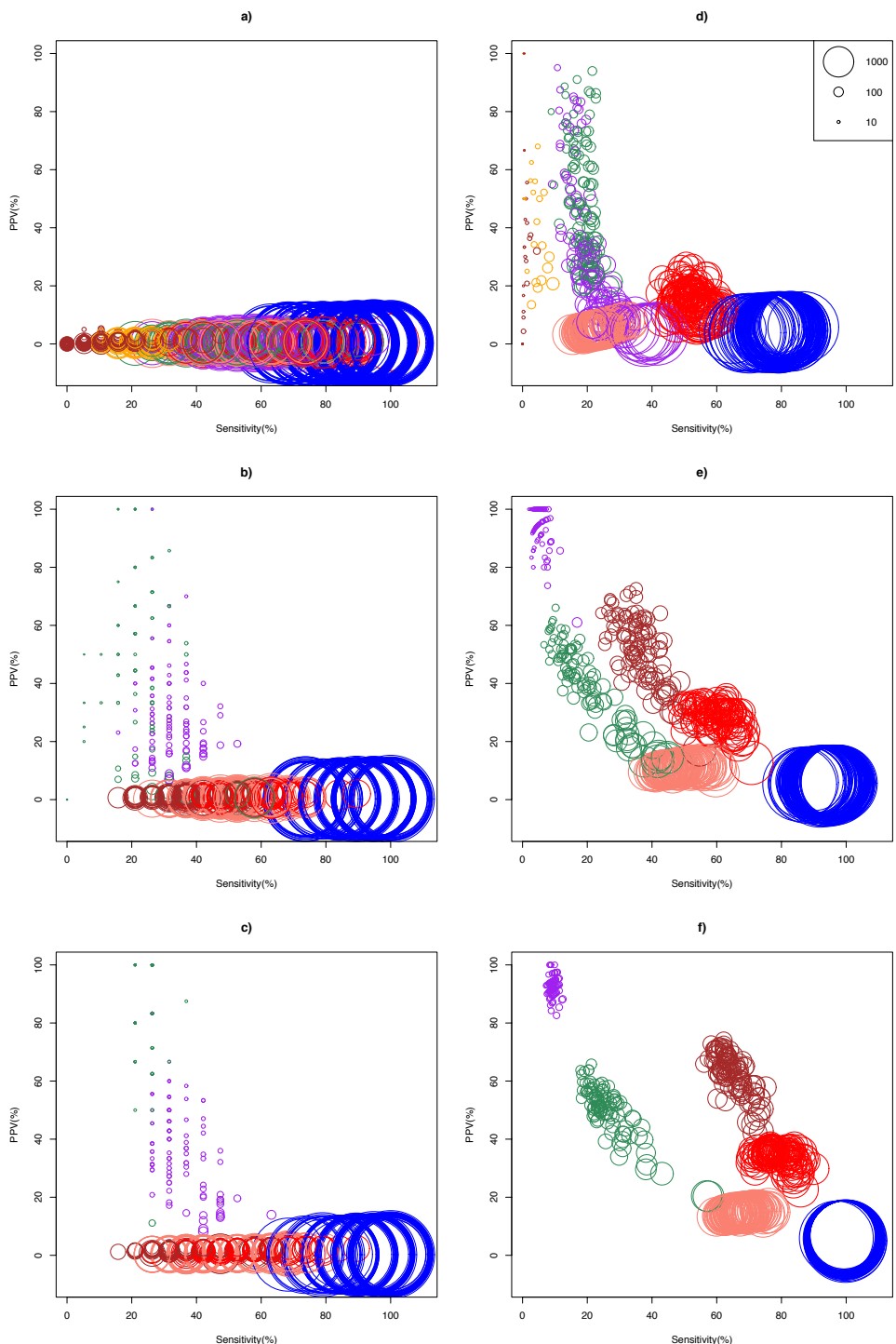

**Figure 2 Scatter plots of PPV against sensitivity.** The $n = 1, 3, 6$ scenarios are given in (A, B, C) for the Cheung data set, and (D, E, F) for the Bottomly data set, respectively. The diameter of a circle is proportional to the DEG set size (scale provided in Fig. 2D). Color legend: blue (*Z*-test), pink (DESeq), red (edgeR), brown (DESeq2), purple (GFOLD), green (NOISeq), orange (ASC). For $n = 10$ in the Cheung data set, see Fig. S3.

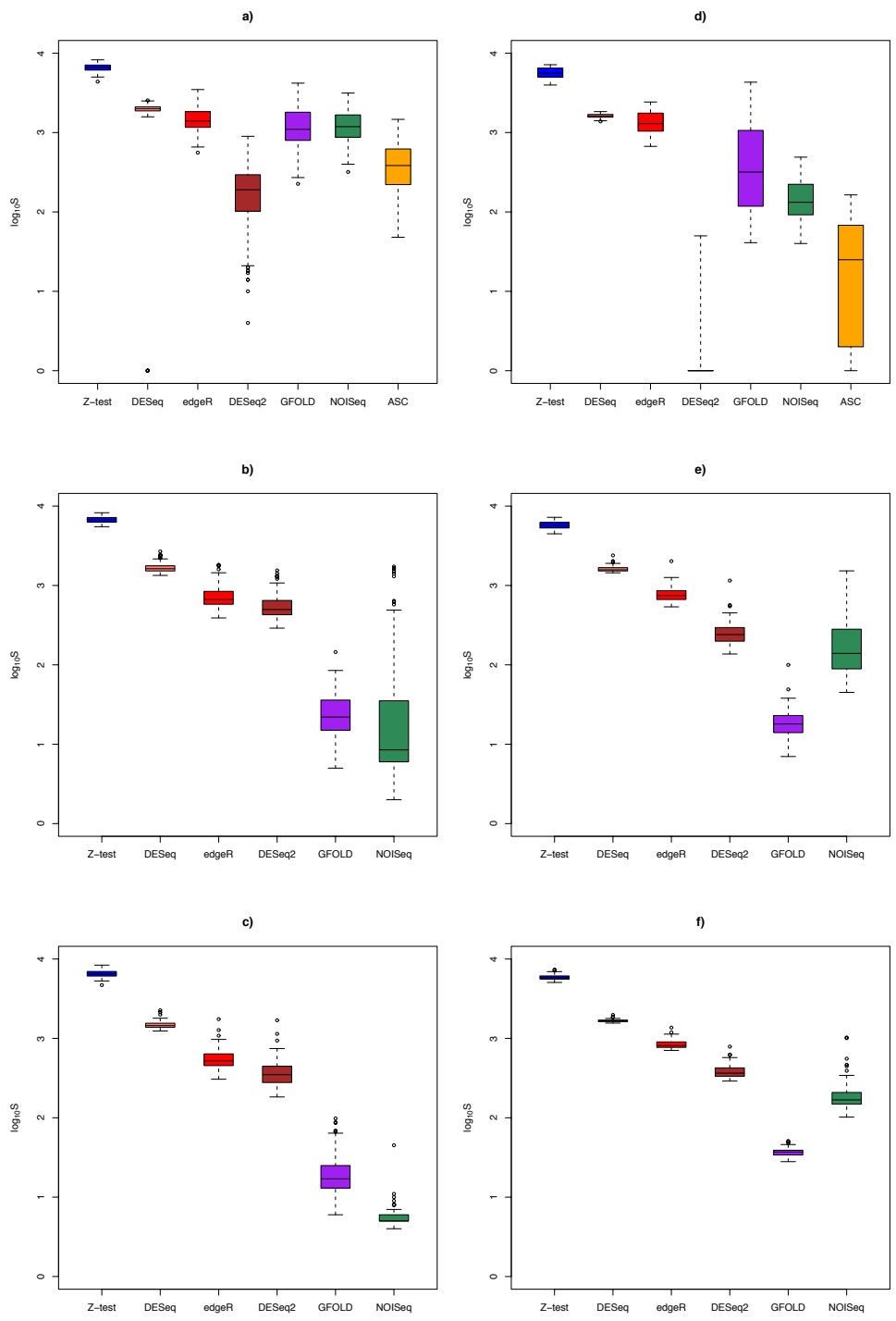

**Figure 3 Box plots of distribution of DEG set size (in log₁₀ scale) by method.** The $n = 1, 3, 6$ scenarios are given in (A, B, C) for the Cheung data set, and (D, E, F) for the Bottomly data set, respectively. Color legend: blue (Z-test), pink (DESeq), red (edgeR), brown (DESeq2), purple (GFOLD), green (NOISeq), orange (ASC). For $n = 10$ in the Cheung data set, see Fig. S4.

**Table 4** Pragmatic DEG call methods for four combinations of biological effect size and replicate size, with suggested applications.

| Replicate size | Biological effect size | |
| --- | --- | --- |
| | Mild | Strong |
| 1 | Nothing works | GFOLD[v], NOISeq[v] |
| 3+ | GFOLD[v], NOISeq[v] | GFOLD[v], DESeq2[s], edgeR[s] |

**Notes.**
v, for validation work; s, for systems biology work.

## Optimality requires a context

The current results suggest that unreplicated RNA-seq experiments, which are still very common among underfunded labs working with non-model organisms, may be a cost-effective way to generate candidate DEG with reasonable likelihood of being validated, provided that the underlying biological effect size is strong. Thus, for unreplicated RNA-seq experiments with phenotype classes such as those associated with pathogenic challenge and physico-chemical stress, we expect DEG called using NOISeq or GFOLD to be good candidates for validation. ASC may also be useful, though it should be noted that it could fail to run for particular combinations of sample pairs, as we found out in the present study. For validation work, GFOLD and NOISeq should be even more efficient once triplicates are available, but further replicate size increase produced only marginal mean PPV gain in the Bottomly data set, suggesting that using more than triplicates is not a cost-effective approach when validation of DEG candidates is the main research goal. When biological effect size is strong, we suggest DESeq2 or edgeR as promising methods to mine DEG for systems biology work, on account of their good PPV and sensitivity balance. However, users should be aware of possibility that shrinkage estimation of dispersion and fold change procedure in DESeq2 may over-correct the initial estimates of these parameters, leading to a DEG set size that is too small, as discovered in the analysis of the Rajkumar data set (Table 3).

Research programs focusing on the investigation of weak or modest biological effect sizes must have replicates, use NOISeq or GFOLD for DEG calling, and then to restrict the research goal to validation of the DEG candidates. Pursuing a systems biology (e.g., gene set analysis, functional enrichment) direction in such programs is not feasible, since in the Bottomly data set, the DEG set size of both GFOLD and NOISeq at $n = 10$ became too small (below 20).

Table 4 summarizes the recommended DEG call methods and research goals for the combinations of biological effect size and replicate size considered in the present study.

## Transcriptome coverage effect

Transcriptome coverage can be another important source of variation for the observed RNA-seq gene counts (*Sims et al., 2014*). Assuming transcriptome size was approximately equal for human and mouse, relative transcriptome coverage was about three times larger in the Cheung data set (human) compared to the Bottomly data set (mouse). Despite this, detection of DEG remained difficult when biological effect size was mild, suggesting

that the effect of transcriptome coverage on DEG calling was probably marginal in the present study.

## Limitations

In the present study, we justified the use of voom for setting the reference DEG set on the basis of its performance in the Rajkumar data set, which has 115 qPCR-validated genes. Ideally, analyses of additional data sets of this nature would help us better understand the variability of method performance. Unfortunately, RNA-seq data sets that are coupled with extensive qPCR validation results remain uncommon.

The cost of not constructing the reference gene set using a simulation approach was the lost of one degree of freedom in the number of DEG call methods that could be evaluated, since we had to select one of the methods to determine the reference DEG set. Because of this, it may be possible that voom was actually the ideal method for making DEG calls when sufficient replicates are available. Therefore, in practical situations where systems level analysis is desired, one may wish to consider taking the union of DEG set called using voom with that from DESeq2 or edgeR. If the size of the union set is too large, one may consider taking the intersection set instead to obtain a smaller, but higher confidence DEG set (*Zhang et al., 2014*).

## Future prospects

Many biologists have difficulty publishing results of RNA-seq experiments with no or few biological replicates. Despite including qPCR validation results, these studies are often dismissed by reviewers simply on grounds of 'not having enough sample size.' This stand is unnecessarily dogmatic, and does not take into account that some particular combinations in the trinity of replicate size-effect size-call method can potentially yield biologically meaningful results, as shown in the present study.

It is gradually being appreciated that RNA-seq analysis is a complex analysis that needs to address the numerous sources of variation from library preparation to bioinformatic processing (*Kratz & Carninci, 2014*) to yield an interpretable result. As a corollary, we suggest that one-size-fits-all pipelines for RNA-seq analysis commonly adopted by bioinformatics service providers should not be expected to always yield the most optimal set of DEG. There is a certainly a need for greater consultation between scientist and the bioinformatician to fine-tune pipelines by taking into account interactions in the replicate size-effect size-call method trinity.

As more high-quality RNA-seq experimental data continue to accrue in public databases, a better understanding of the anticipated behavior of various DEG calling methods under different biological and replicate size scenarios should gradually emerge from systematic comparison studies such as the current one. A complete dummy's guide to RNA-seq differential gene expression analysis may not be too far ahead in the future.

## CONCLUSIONS

In RNA-seq experiments, biological effect size is an important determinant of whether a research program at the individual gene or systems level would yield the most biological

insight. When it is expected to be mild, RNA-seq experiments should primarily aim at mining DEG for validation purpose, using at least triplicates and either NOISeq or GFOLD for DEG calling. Moreover, systems level analysis remains difficult as none of the methods considered presently showed satisfactory sensitivity and positive predictive value performance. When strong biological effect size is expected, analysis of unreplicated experiments using GFOLD or NOISeq can yield DEG candidates with optimistic validation prospects. The use of triplicates or more not only improves the statistical power of DEG call methods (*Liu, Zhou & White, 2014*), but also unlocks the analytical potential of RNA-seq data sets. Thus, users can apply GFOLD to pinpoint a DEG set for targeted qPCR validation, and simultaneously implement DESeq2 or edgeR to identify a DEG set for systems level analysis. Combining results from voom with those from DESeq2 or edgeR may lead to further improvements.

## ACKNOWLEDGEMENTS

We are grateful to Jose M.G. Izarzugaza, Hao Zheng, an anonymous reviewer, and Jaume Bacardit (Academic Editor) for their helpful and constructive comments which resulted in important improvements to the present work.

### Funding

The study was supported by the University of Malaya Research Grant number UMRG RP032D-15AFR to Tsung Fei Khang. The funders had no role in study design, data collection and analysis, decision to publish, or preparation of the manuscript.

### Grant Disclosures

The following grant information was disclosed by the authors:
University of Malaya Research Grant: UMRG RP032D-15AFR.

### Competing Interests

The authors declare there are no competing interests.

### Author Contributions

- Tsung Fei Khang conceived and designed the experiments, analyzed the data, wrote the paper, reviewed drafts of the paper.
- Ching Yee Lau performed the experiments, analyzed the data, prepared figures and/or tables, reviewed drafts of the paper.

### Data Availability

R codes for the computational analyses done are available at http://github.com/tfkhang/rnaseq.

## Supplemental Information

Supplemental information for this article can be found online at http://dx.doi.org/10.7717/peerj.1360#supplemental-information.

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
