# Peer review of "Getting the most out of RNA-seq data analysis"

_PeerJ, doi:10.7717/peerj.1360_

## Round 0.1 · original submission · Major Revisions

· Academic Editor

Major Revisions

Please address carefully all issues raised by the reviewers, especially about the definition of your 'gold standard'. I would strongly suggest to follow reviewer 2's advice and use a dataset for which qPCR validation data exists

·

Basic reporting

The submitted manuscript adheres to current standards in the field and I do not see any main reason that should prohibit the review of this article. The text is written clearly in proper English.

The quantitative analysis of the relationship between biological effect, number of duplicates and DEG calling performance is sound and of interest to the scientific community.

Experimental design

My main concern regards the validity of the gold standard for comparison. Currently, the authors use a gold standard that is based on predictions where “at least 70% are expected to be real”. The presented results, then, do not evaluate how much the predictors perform on predicting DEG. Contrarily, they assess the degree of correlation between their predictions and voom’s. The discussion of this aspect is very vague in the manuscript. The author should provide a solid argumentation of the validity of the voom gold-standard. If experimental validation is included, it might turn out that voom itself is the methodology of choice for the prediction of DEG.

Validity of the findings

Following the previous comment, the authors should justify the validity of this assumption “Assuming that at most half of them were false positives, at least 70% of the voom-called DEG were expected to be real.” Otherwise, they might consider providing the results as a range that considers (all correct, all false positives).

Figure 2 is difficult to follow. I would suggest that the information regarding the number of replicates and the calling method is incorporated in the scatterplots as a legend. The number of DEGs predicted might be encoded as a color whereas the predictor could be a shape, the diameter is difficult to interpret. Also, it might be benefitial for the reader to include n=10 currently provided as Supplementary. I hope this would improve the interpretation of the results.

Additional comments

What I would consider minor comments:
- Closing parenthesis in “Supplemental Material Table 1). The remainder …”
- To facilitate the interpretation of the results I would refer to the datasets as “mild response” and “strong response” instead of the study names as these terms relate directly to the conclusions drawn.
- In the abstract, the authors should specify that PPV refers to positive predictive value before using the acronym.

Reviewer 2 ·

Basic reporting

- Introduction, 2nd paragraph, 3rd line: “thorugh” should be changed to “through”.
- When considering replicates, Z-test is also applied. How are the proportions for a given gene and phenotype class computed? As a mean of proportions? Please, be more specific on this.
- Criteria for differential expression, 1st paragraph: I understand that DE calls are selected according to p-value (adjusted p-value I guess) and fold-change but it is not clear for me the reason to add the explanation on x and y (from “The product of y>2” until the end of the paragraph, and therefore the definition of x and y). I think that it is enough if the threshold for p-value and fold-change are indicated.
- Criteria for differential expression, last paragraph: The first sentence is incomplete (“a fold change of” ??). What is then the GFOLD value?
- I missed a short description of the differential expression methods to be compared. Most readers may know about edgeR or DESeq and DESeq2, but perhaps they do not know the procedure the rest of methods use to compute differential expression. A brief explanation would be enough so they do not have to go to the original papers.
- Benchmarking, 4th paragraph: “The exception is DESeq2, which specifically requires raw counts instead of normalized count for analysis”. Is this only true for DESeq2? As far as I know, edgeR also requires raw counts, except the authors mean that edgeR accepts other normalization or scaling factors.
- Transcriptome coverage effect: Why are the authors discussing about similarities of transcriptomes between human and rat? The Bottomly data is from mouse, not rat. Furthermore, the number of transcripts in human is approximately twice the number of mouse transcripts. Could you elaborate more on this?
- In general, I think that the authors should state more clearly that to perform a real statistical inference analysis, biological replicates are needed. Otherwise the results obtained cannot be generalized to the population and validation is essential (as they already mention). They could also include references that sustain the need of triplicates as many studies are recently claiming.

Experimental design

The manuscript provides a comparison of six differential expression methods for RNA-seq data on two publicly available datasets. The authors use two performance parameters: sensitivity and positive predictive value. To compute them, they define a gold-standard set of DEGs by applying voom transformation and limma method. In my opinion, it is arguable that the set of DEGs considered as a gold-standard can be defined from the results of a particular DE method. All methods are subject to declare false positives or negatives, and using this as a gold standard may produce misleading conclusions. The results of the study would have more impact if at least an additional RNA-seq dataset with qPCR validation were included to corroborate at least some of these conclusions.

I also have some concerns about the choice of the dataset with strong biological effect. Providing the set of truly differentially expressed genes is correct, there are only 362 DEGs (less than 3% if we consider only the genes with counts). According to my experience and to the literature, I do not think this can be considered as a “strong effect”. I agree that simulation experiments may have some limitations but at least they can control better the size of the effects, which by the way, it is not related only with the number of DEGs but also to the magnitude of change between phenotype classes. Could the authors provide more information about this magnitude? And a discussion on my comment?

The authors are right when considering the importance of the biological effect size when calling DEGs. However, other factor, which is also important in any statistical analysis such as DE, is the biological variability within the same experimental group (phenotype class) when biological replicates are available. Could the authors discuss in the manuscript if the biological variability within groups is the same for both datasets, groups of replicates compared, etc. and if it is affecting some how the results of any of the compared methods. It could be useful for the readers because it is difficult to estimate the size effect in a given experiment, but variability can easily computed.

Validity of the findings

The novelty of this comparison study is that it focuses in the case of non-replicated data, which is still quite common in RNA-seq. As mentioned before, the validity of the results and conclusions from the DE methods comparison may depend on the chosen data sets. It is difficult to generalize the conclusions with only two examples so the authors should carefully analyze the impact of other factors such as the magnitude of change between classes or the variability within classes. They should also provide another example in which a more widely accepted gold standard set of DEGs such as qPCR was available to reinforce their conclusions.

·

Basic reporting

In this manuscript, the authors investigated seven methods in detecting differentially expressed genes (DEGs) using RNA-seq data. These methods are, namely, GFOLD, ASC, NOISeq, edgeR, DESeq, DESeq2, and Z-test. These methods were applied to two sets of publicly available RNA-seq data sets: (1) the Bottomly data set of mice brain striatum tissues; (2) the Cheung data set of immortalized human B cells. The Bottomly data set includes two mice strains, whose difference represents a case of strong biological effect. The Cheung data set puts emphasis on gender difference, as an example of mild biological effect. Comparisons based on sensitivity and positive predictive values were implemented, and the sizes of identified DEGs were also presented. Through such investigations, the authors reached recommendations for RNA-seq analysis in general, with respect to the impact of sample sizes (biological replicates) and biological responses.

Experimental design

As a method comparison study, the design of this work is relatively straightforward. While the rationales of choosing the data sets are explicitly explained, the reasons of including such seven methods are not interpreted in detail. In the section of Introduction, and Materials and Methods, only the applicability of NIOSeq, GFOLD, and ASC with respect to biological replication (whether an analysis using one sample per phenotype can be conducted) was provided. Whether these methods in comparison are state of the art for RNA-seq analysis has not been discussed.

In the simulated unreplicated experiments, 27 and 124 pairs of samples were chosen for two data sets, respectively, because the ASC package encountered problems in dealing with the remaining pairs. This strategy is not preferred, for this manner of subset selection may introduce bias in comparing performance. It is better to run all combinations (or also randomly select 100 instances, like for n=3, 6, and 10) for seven methods, and to point out the suitability of individual methods.

Validity of the findings

The 'gold standard' used in this work for confirming DEGs is the output from implementing voom algorithm. Although the voom algorithm transforms the count data into a microarray-like data type so that the mature limma analysis could be applied, it is not convincing that the results based on voom are closer to the real situations. In fact, if voom by default outperforms the current RNA-seq analysis methods (such as the seven ones in comparison), this work of comparing these methods is not meaningful. Therefore, the statement of 'golden standard' might need rephrase or further justification, and the following implicated presentations need corresponding adjustments as well.

The comparisons among the involved methods are presented mainly through Figures 2 and 3, and their corresponding interpretations in the section of Results and Discussion. While the numbers on the top of Page 5 are systematically reported, it is difficult to link them to the counterparts in Figure 2, especially when the figure has six panels and each panel covers seven colors. It would help if the expression of xx +/- xx % is followed by (Panel x, Figure 2). Based on the current setting, the numbers of circles in Panel (a) and (d) of Figure 2 are different from the remaining four, so listing the numbers of circles could avoid potential confusion. Because the size of identified DEGs is shown via the diameter of the corresponding circle, an illustrative legend could better convey the information. Meanwhile, the wording "... the mean size of DEG called was very small (6+/-11.6%) " (second paragraph on Page 5) is rather confusing. The size of DEGs should be an integer, rather than some percentage.

Due to the big gap of sizes of selected DEGs, Figure 3 has to use log scale to show a normal box plot. However, it is not straightforward to map the numbers back in originally linear scale. It should largely contribute to the presentation if there is a table showing the summary statistics for the DEG sizes in the linear scale.

Additional comments

There are a few minor issues.

The manuscript context says there are 24 males and 17 females in Cheung data set on Page 3, but the caption of Figure 1 says there are 17 males and 24 females.

In the section of Introduction on Page 2, 'through' is misspelled as 'thorugh', and the past tense of 'offer' should be 'offered' instead of 'offerred'.

In the section of Conclusion on Page 6, 'Moreover' is misspelled as 'Morever'.

---

## Round 0.2 · accepted · Accept

· Academic Editor

Accept

The concerns of the reviewers were thoroughly addressed. Please, proofread your manuscript before the production phase, following the suggestions of reviewer 1.

·

Basic reporting

Although the authors have improved the text from the previous version, the use of English should be revised in detail. Some sentences are convolute and difficult to follow. There are some incomplete sentences and spelling mistakes. Some examples:
Introduction
- Format "de novo” in italics as it originates from Latin. (… numerous de novo transcriptome assembly tools …)
- Consider revising the punctuation of the following sentence “Despite microarrays’ analytical maturity … future gene expression studies".
Materials and methods
- The relative performance of various subsets *of* these methods …
- Method for constructing *a* reliable reference DEG set

Experimental design

R1.1 The authors have improved their dissertation on the validity of the voom dataset as reference for comparison with the other methodologies. Although the benchmark with the Rajkumar dataset only considers 2 of the evaluated methodologies, voom results are better than those of edgeR. Still, the manuscript is based on indirect predictions and could be improved in the context of experimentally validated datasets as the other reviewers also highlight.

Validity of the findings

No Comments

Additional comments

No additional comments

Reviewer 2 ·

Basic reporting

Ok

Experimental design

Ok

Validity of the findings

Ok

·

Basic reporting

No extra comments in addition to those provided during the previous review.

Experimental design

No extra comments in addition to those provided during the previous review.

Validity of the findings

No extra comments in addition to those provided during the previous review.

Additional comments

The authors provided responsive replies to my comments. A brief review of currently available RNA-seq data analysis methods was presented, and the rationale of selecting involved approaches in comparisons in this work was provided. The authors updated the numerical investigation upon my comments in using the Cheung and Bottomly data sets to avoid possible selection bias introduced by the inapplicability of ASC package. This update should strengthen the presentation.

The challenge of lack of empirical evidence to justify the findings still exists. The doubt of using voom transformation together with limma method to determine differentially expressed genes is raised by all three reviewers. To address this concern, the authors referred to the work of Rajkumar et al. (2015) to defend the effectiveness of voom approach. Meanwhile, the authors circumvented the original question by claiming that voom approach is one of the most effective methods, rather than the "gold standard". A subsection of "Limitations" was added to the section of "Results and Discussion" to explain the dilemma of using the result from one method as reference. Although the ideal practice is still to put voom approach into the competition with other methods, and to use qPCR validation as the evaluation criterion, the strategy adopted by the authors appears reasonable, given the potential shortage of related qPCR-validated outcomes.